# Autotoxicity of Endogenous Organic Acid Stress in Two *Ganoderma lucidum* Cultivars

**DOI:** 10.3390/molecules27196734

**Published:** 2022-10-09

**Authors:** Pan Zou, Yongze Guo, Shu Ding, Zhaowei Song, Hanyuan Cui, Yue Zhang, Zhijun Zhang, Xiaoming Chen

**Affiliations:** Tianjin Academy of Agricultural Sciences, Tianjin 300380, China

**Keywords:** *Ganoderma lucidum*, organic acids, autotoxicity, cropping obstacle

## Abstract

*Ganoderma lucidum* has been used as a rare medical mushroom for centuries in China, due to its health-promoting properties. Successive cropping obstacles are common in the cultivation of *G. lucidum*, although the remaining nutrients in the germ substrate are sufficient for a second fruiting. Here, we aimed to study the metabolite profile of *G. lucidum* *via* nontargeted metabonomic technology. Metabonomic data revealed that organic acids played an important role in the cropping obstacles of *G. lucidum*, which is accordance with the pH decrease in the germ substrate. A Kyoto encyclopedia of genes and genomes (KEGG) enrichment analysis indicated that most differential acids participated in the metabolic pathways. Five acids were all significantly upregulated by two MS with high energy (MS^E^) modes in two cultivars, among which 5-hydroxy-2-oxo-4-ureido-2,5-dihydro-1*H*-imidazole-5-carboxylic acid is also involved in purine metabolism regulation and microbial metabolism in diverse environments. Taken together, this work illustrated the organic acid stress generated by *G. lucidum*, which formed the autotoxicity feedback, and resulted in cropping obstacles. Determining the cause of the cropping obstacles in *G. lucidum* will promote the utilization rate of fungus substrate to realize the sustainable use of this resource.

## 1. Introduction

Most edible mushrooms can produce a qualified fruiting body at least two consecutive times under proper artificial conditions, such as *Pleurotus ostreatus*, *Flammulina velutipes*, *Coprinus comatu*, etc. Other edible fungi only produce qualified fruit the first time, and then tend to consecutively produce defective fruits [1]. This is known as a successive cropping obstacle, which is common during the culture of *G. lucidum*. *G. lucidum* is one of the most typical varieties, with abundant bioactive components [2,3], including triterpenoids, polysaccharides, nucleosides, alkaloids, ganoderic acid, etc. The functional compounds’ immunoregulation, antitumor, neuroprotection, hepatoprotection, antioxidation, anti-radiation, and anti-mutation properties have been proven [4,5,6] since the discovery of wall-breaking technology. The medical mushroom has shown promising potential for pharmacological and economic applications. However, successive cropping obstacles limit its cultivation promotion, as only the first fruit of *G. lucidum* has large output and contains plentiful bioactive substances; the fruiting body, output, and bioactive substances from the second fruiting are unsatisfactory [7]. It has been considered that the successive cropping obstacles in *G. lucidum* are mainly related to environmental conditions and nutrients in cultivation materials. Although the remaining nutrients in the germ substrate are enough for continuous fruiting, the remaining cultivated substances cannot be applied to growing homologous edible fungi again, unless the germ substrate or covering soil are exposed to sun, rain, and decay over two years. This is a passive way to change the cultivation environment and supplementation of germ nutrients, which will also increase the cost, instead of enhancing the utilization of cultivation substances as an alternative.

The continuous cropping obstacle in edible fungi is similar to that in crops and vegetables, which is closely related to root exudate dispersion and accumulation in soil [8,9,10,11]. Fungi metabolites inhibit its second fruiting when exudates accumulate to a certain amount. Recently, we found that the germ substrate pH significantly decreased from 7.50 to 4.74 after the first fruiting, indicating that the mycelium of *G. lucidum* secreted organic acids or other metabolites into cultivated substances, and resulted in the subsequent fruiting body becoming defective. Moreover, the ethanol extract and water extract of the germ substrate after the first fruiting had an inhibitory effect on the growth of homologous *G. lucidum*, and the inhibition was positively correlated with extract doses (unpublished data). The germ substrate contains almost mycelium-secreted organic acids; meanwhile, the accumulated organic acid may inhibit feedback on its own growth. Continuous cropping increases the accumulation of inhibitors and changes the soil microenvironment. Therefore, we assumed that the organic acid exudate may be the main reason for the failure of continuous cropping.

To the best of our knowledge, there have been few reports on the feedback inhibition of *G. lucidum*. In this study, we compared the changes in organic acids at different *G. lucidum* growth stages using non-targeted metabonomic technology. Then, we determined the differential organic acids and related pathways, which were used to analyze the promising autotoxicity mechanism. The metabolite feedback inhibition study on edible mushrooms is helpful when screening inhibiting components and inhibition scavengers, which will overcome successive cropping obstacles, make full use of cultivation substances, and improve the biological efficiency of fungi culture.

## 2. Results and Discussion

### 2.1. pH Variation of Germ Substrates during Cultivation

Figure 1 shows the pH changes in substrates from inoculation to a mature fruiting body. Arrows point to important growth stages of *G. lucidum*, including primordium at Day 8, pileus at Day 18, fruiting initiation at Day 28, and spore ejection at Day 42. As seen in Figure 1, pH decreases can be observed before every stage of growth point, especially before spore ejection (Day 42). The pH of L09-1 (5.36) decreased to 4.63 from the inoculation to the end of first fruiting, and the pH of L09-2 showed little change from 4.46 to 4.3. The transcriptional dynamics of genes during sporulation in *G. lucidum* revealed that the differential expressed genes involved in the carbohydrate metabolic process were upregulated at the later stage of sporulation, and the accumulation of specific carbohydrates may serve as energy sources for further development [12]. The decreased pH of the germ substrates indicated the increase in some organic acid compounds. Organic acids can modify the physical and chemical properties of substrates, and influence nutrient absorption; therefore, organic acids play an important ecological role in the growth of *G. lucidum*. In addition, organic acids in medical mushrooms are potential candidates as antifungal agents, resulting in autotoxicity in the second fruiting [1,13,14]. Therefore, we focused on the organic acid and derivative changes in germ substrate affected by *G. lucidum* through non-targeted metabolomics.

### 2.2. Acquisition of Mass Spectrum Data

Since quality control (QC) samples are prepared with mixed samples, the differences in the varieties and abundances of metabolites in QC samples should be within the random error. Therefore, the consistency of QC sample results can be used as an important method to evaluate laboratory automation and quality management. In this study, samples from different groups were arranged in a random order, and a QC sample was inserted every ten samples. In Figure 2, the heat map of the QC samples’ correlation was entirely above 0.9, indicating that the data obtained from HPLC-QTOF-MS were valid.

### 2.3. Comparison of Organic Acids and Derivatives between L09-1 and L09-2

The separation and identification of organic acids from the germ substrate was performed using UPLC-QTOF-MS^E^. MS^E^ is an MS with high energy that can collect ion fragments ranging from low to high energy. Figure 3 shows an example of chromatograms with marked retention times of peaks obtained in positive and negative modes. According to the positive MS^E^ mode by UPLC-QTOF-MS, 3521 metabolites in the two varieties were screened through qualitative and quantitative analysis, among which 1007 metabolites were significantly different. A total of 724 compounds out of 3521 metabolites belonged to organic acids and derivatives. Appendix A lists 194 significantly upregulated and downregulated organic acids and derivatives in both species. There were 119 upregulated compounds, while the others were downregulated.

For the negative MS^E^ mode, there were 43 differential metabolites in total, among which there were 17 organic acids and derivatives. As shown in Appendix A, nine organic acids were all upregulated in two *G. lucidum*, and only the content of C19:1 (CIS-10) acid was downregulated, compared to that of the inoculation. The reference showed that organic acids are closely related to the synthesis and catabolism of the cell wall [15,16]. The cell wall is the first barrier for external ions entering the protoplasm, which plays an important role in the resistance to external ion stress [17]. The feedback of organic acids may inhibit the synthesis of the cell wall during the second fruiting, thus producing defective *G. lucidum*.

In Figure 4, the KEGG pathway enrichment analysis indicates that most of the differential acids participate in the metabolic pathways, which is in keeping with the finding of Cai and colleagues [12]. There were 22 acids in positive mode and 1 in negative mode (map01100, seen in Appendix A). Table 1 lists five acids, which appeared in two MS^E^ modes, and all were significantly upregulated in the two cultivars. The differential compound 5-hydroxy-2-oxo-4-ureido-2,5-dihydro-1*H*-imidazole-5-carboxylic acid—which occurs in both MS^E^ modes and cultivars—is also involved in both purine metabolism regulation (map00230, seen in Appendix A) and microbial metabolisms in diverse environments (map01120, seen in Appendix A). However, other compounds in Table 1 were not found in the related KEGG pathway. Moreover, 13 differential organic acids were involved in the biosynthesis of secondary metabolites (map01110, in Appendix A), 5 organic acids were involved in the biosynthesis of amino acids (map01230, in Appendix A) and 3 organic acids were involved in lysine degradation (map00310, in Appendix A) in positive ion mode. Other than that, either two or one differential compounds in positive ion mode appeared in the corresponding KEGG pathway (data not shown). The KEGG pathways suggest that these metabolic processes may be important in connecting edible mushrooms’ secondary metabolic networks [18].

*G. lucidum* is an important edible fungus with medical value [19]. Improvements in the yield and utilization rate of the germ substrate is helpful to lower the planting cost and to avoid wasting resources. However, most reports have been concerned with evaluations of its medical activity and analyses of its functional ingredients, but paid little attention to the metabolite-autotoxicity-induced successive cropping obstacles [20]. In our study, we found that organic acids and derivatives account for the majority of all differential metabolites. Most differential organic acids were upregulated in two *G. lucidum* cultivars, which is in accordance with the decrease in the pH of the germ substrate observed during cultivation (seen in Figure 1). In the study of Xie et al., most triterpenoids, including ganoderic acid, guanidylic acid, and lucidenic acid, were upregulated in the fruiting body stage, and this may be positively correlated with the increase in the organic acid content in the substrate [5]. Organic acids can preserve the quality of fruit and defend against abiotic stress [21,22]. The increased organic acid content may offer a reasonable explanation for the fact that the remaining nutrients of the germ substrate cannot be reapplied to grow qualified *G. lucidum*. The content of organic acids represent the metabolic status, and a higher level of organic acids reflect an organism’s ability to survive and maintain its basal metabolism [23]. The accumulation of organic acids increases energy production and amino acid content and better maintains the osmoregulation of cells, thus enhancing the tolerance to stress [22,24]. Mycelia secrete organic acids into the germ substrate and change the environmental state. Organic acids may negatively regulate the growth in the fruiting body by participating in metabolic pathways, purine metabolism regulation, and microbial metabolism in diverse environments or other pathways, thus forming the autotoxicity feedback. In addition, the substrate bacterial community structure may be affected by the pH change [25,26], and may in turn influence the growth of *G. lucidum*, which will be considered in a further study of ours.

## 3. Materials and Methods

### 3.1. Strains and Inoculation

Strains of *G. lucidum* were bred by our lab and were named L09-1 and L09-2. The germ substrate consisted of 78% wood chips, 20% wheat bran, 1% lime, and 1% gypsum; they were then mixed with equivalent water [27,28,29]. Polypropylene bags containing 2 kg of the substrate were ready for mycelium inoculation after autoclaving at 121 ℃ for 40 min. Next, L09 and L09-2 were planted on solid substrates. After inoculation, germ barns were collected every 48 h until the end of first fruiting. A total of seven samples were collected during the assay. The substrate samples were dissolved in hot water and filtered. Supernatants were ready for subsequent analysis. The pH values of each sample were determined. All other reagents were of the highest available quality (Sigma-Aldrich Co. Ltd., Burlington, MA, USA).

### 3.2. Sample Preparations

A total of 100 mg germ substrate sample was ground with liquid nitrogen, and vortexed with 300 μL of 80% methanol (Sigma-Aldrich Co. Ltd., Burlington, MA, USA). The mixture was ground by a tissue crusher (Zhengqiao Instrument Company, Shanghai, China) at 45 Hz for 4 min, and then treated with ultrasonication in an ice water bath for 5 min. Samples were kept at −20 ℃ for 30 min, and centrifuged at 14,000× *g* for 30 min. The supernatant was filtered ready for further measurement. The quality control (QC) sample was prepared by drawing 10 μL of supernatants from all abovementioned samples and then mixing them up. Samples from different groups were arranged in a random order, and a QC sample was inserted every ten samples.

### 3.3. UPLC-MS/MS Data Acquisition

Samples were determined by Waters Acqquity UPLC equipped with Waters Vion IMS QTOF (Waters Co. Ltd., Taunton, MA, USA). A 5 μL sample was loaded and separated on a Waters ACQUITY UPLC HSS T3 C18 column (2.1 × 100 mm and particle size of 1.8 µm) with 0.1% formic acid aqueous solution (eluent A) and 0.1% formic acid acetonitrile solution (eluent B) in gradient elution mode at a rate of 0.25 mL/min. The Elution program was set as follows: from 98% A at 0 min to 70% A at 5 min; constant 50% A from 7 to 16 min; from 10% A at 16 min to 2% A at 17 min; constant 2% A from 17 min to 20 min. The column temperature was kept at 25 ℃.

The MS detection conditions were as follows: the desolvation gas N_2_ flow was 800 L/h at 500 ℃, and the flow rate of cone gas is 50 L/h. The capillary voltages were set at 2.00 KV and 1.20 KV for positive and negative MS^E^ modes, respectively, and separately. Scan range was from 100 to 1200 *m*/*z*, and scan time was 0.3 s.

Samples were analyzed in random sequence to avoid the influence caused by fluctuations in instrumental signals. QC samples were inserted every ten samples to evaluate the stability of the system and the data reliability.

### 3.4. Processing and Statistical Analysis of MS Spectra Data

The primary data were imported from QTOF-MS to Progenesis QI 2.3 (PQI) software (Waters Co. Ltd., Taunton, MA, USA), and the data were aligned within 0.2 min of retention time deviation with 5 ppm of quality deviation. Peaks were selected according to a coefficient of variance (CV) of 30%, a signal-to-noise ratio of 3, and a minimum signal of 10^5^ in order to provide more accurate identification. Meanwhile, peak area was quantified and target ions were integrated. Organic acids and derivates were predicted by the exact mass of potential difference *m*/*z*. Possible molecular formulas were predicted by comparing ion peaks and fragment ions with the use of databases, which included METLIN, the Human Metabolome database, and ChemSpider. The quantitative results were normalized, and a labeled matrix was produced and recognized [30]. The mass spectra with variable importance in projection (VIP) above 1 were selected in this study, and a non-parametric *t* test with a critical *p* value of 0.05 was applied to further determine whether a significant difference existed between at least two groups for each organic acid and derivate.

## 4. Conclusions

This study is the first exploratory work to determine the cropping obstacle of *G. lucidum*. The metabolic data were analyzed using UPLC-QTOF-MS^E^. Most of the differential metabolites were organic acids and derivatives. Moreover, 119 out of 194 and 9 out of 17 differential organic acids were upregulated in both positive and negative ion modes, respectively. The increased contents of organic acids were consistent with the pH decrease during the whole cultivation. KEGG pathway enrichment analysis revealed that differential compounds were mainly involved in the metabolic pathways. Additionally, we found that the organic acid-induced stress may inhibit the second fruiting of *G. lucidum*. In the next step, we will focus on studying the relationship between successive cropping obstacles and differentially expressed genes and protein in order to find reasonable solutions, or provide novel breeding ideas, for *G. lucidum*. Moreover, the functional nutrients in *G. lucidum* are also important parameters for evaluating its quality, which may be affected by organic acid stress, and will be explored in the future.

## Figures and Tables

**Figure 1 molecules-27-06734-f001:**
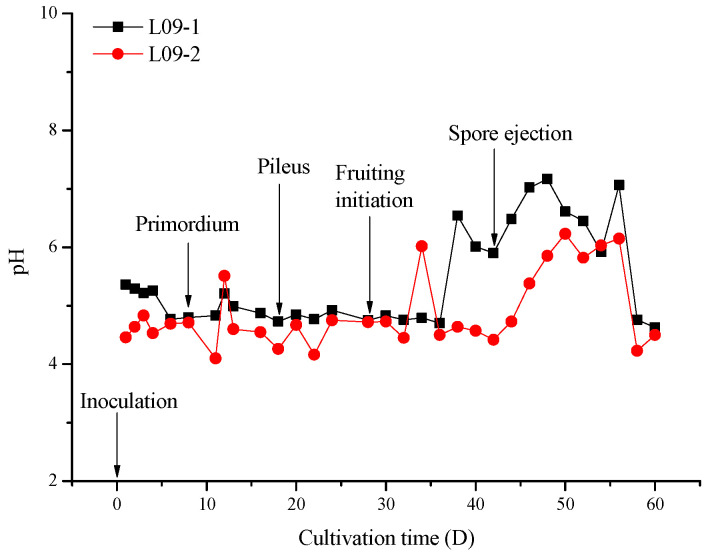
pH values of germ substrates of two *G. lucidum* from inoculation to fruiting. Arrows point to several important growth stages. The pH decreased before every stage of growth point, especially before spore ejection. The pH of L09-1 (5.36) decreased to 4.63 from the inoculation to the end of first fruiting, and the pH of L09-2 showed little change from 4.46 to 4.3.

**Figure 2 molecules-27-06734-f002:**
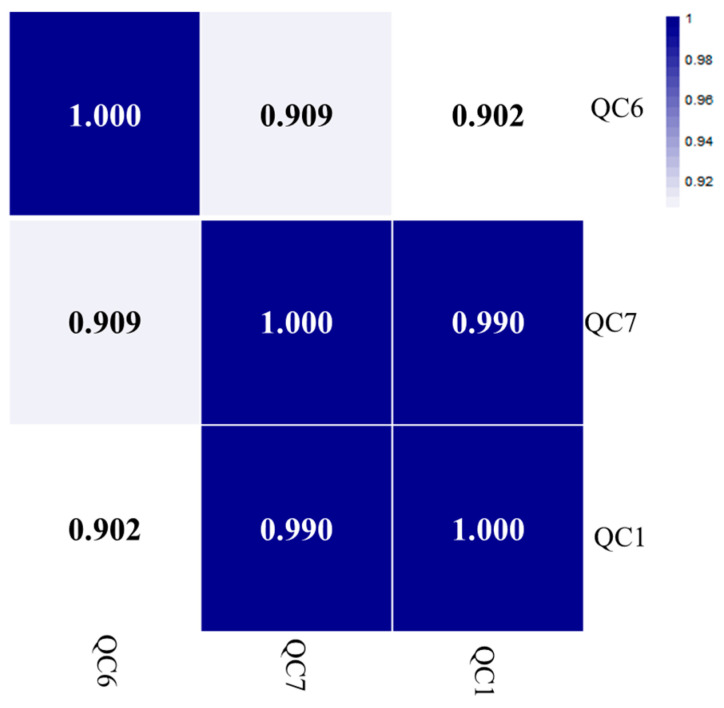
Correlation analysis of QC samples. QC samples were a mixture of all tested samples, used to evaluate instrument stability. The correlation was entirely above 0.9, indicating that the data were valid.

**Figure 3 molecules-27-06734-f003:**
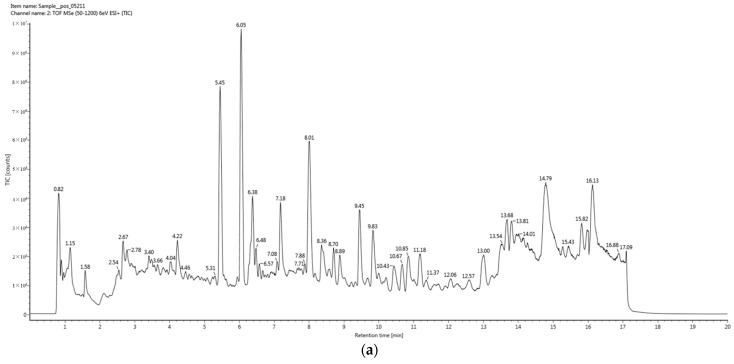
Example of total obtained ion chromatogram of L09-1 germ substrate sample using UPLC-QTOF-MS^E^ in positive mode (**a**) and negative mode (**b**).

**Figure 4 molecules-27-06734-f004:**
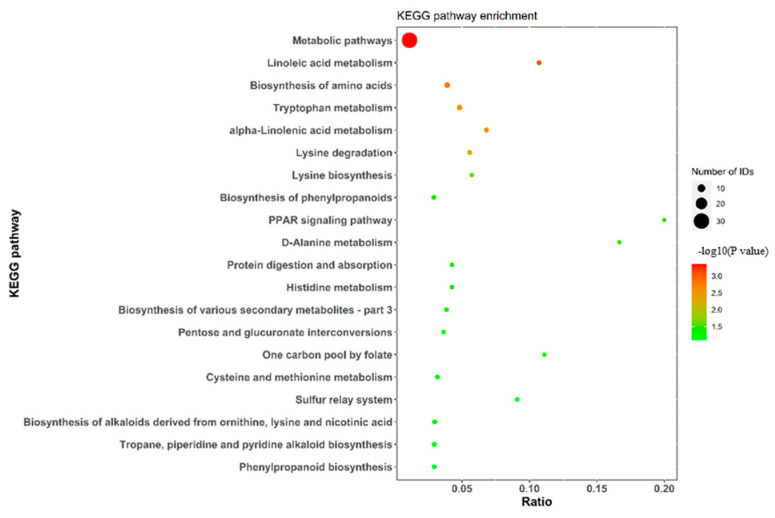
KEGG pathway enrichment analysis of differential metabolites of two cultivars.

**Table 1 molecules-27-06734-t001:** Organic acids and derivatives in the germ substrates of two cultivars using two MS^E^ modes.

No.	Differential Compounds	Content Change
1	5-hydroxy-2-oxo-4-ureido-2,5-dihydro-1*H*-imidazole-5-carboxylic acid	UP
2	malonic acid	UP
3	2-(2-Carboxyethyl)-5-hydroxyphenyl hexopyranosiduronic acid	UP
4	3-C-Carboxy-2-deoxy-4-O-[(2E)-3-(4-hydroxyphenyl)-2-propenoyl]pentaric acid	UP
5	2-Deoxy-4-O-[(2E)-3-(4-hydroxyphenyl)-2-propenoyl]pentaric acid	UP

## Data Availability

The data used to support the findings of this study are available from the corresponding author (Xiaoming Chen, chxm001@126.com) upon request.

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
