# Peer review of "Autotoxicity of Endogenous Organic Acid Stress in Two Ganoderma lucidum Cultivars"

_molecules, 2022, doi:10.3390/molecules27196734_

Round 1

Reviewer 1 Report

At line 10, abbreviate the genus between parenthesis is not necessary. On the other hand, several abbreviations should be better explained, such as MSE and KEGG as Molecules presents a very wide public.

The manuscript needs to be better discussed.

Data is not clear. None of the chromatograms has a clear origin, nor the Tables.

The results at S1 and S2 are very confusing. It seems that authors directly took the proposals from automatic spectrotec and use them, without evaluating the real possibility of those substances being present at natural extracts, such as fluorinated sterols. Several of them details the E,Z and R,S stereochemistry in a extremely detailed way, not possible without all the standards, that were not used in this kind of metabolomic study. For example: Meprobamate, ramipril, alvimopan, carisoprodol and trandolaprilat are synthetic drugs, the first one is a carbamate derivative used as an anxiolytic and tranquilizer, trandolaprilat is a ACE inhibitor, carisoprodol is a muscular relaxant, and so on. Also, Diazinon is an inseticide and 4-Dodecylbenzenesulfonic acid is a synthetic detergent. This last one could be present, but not as a natural compound, but probably as a waste contamination at the glassware.

Reviewer 2 Report

The authors explore the potential correlation of the metabolite profile of Ganoderma lucidum (G. lucidum) and successful secondary cropping of the plant. That the topic is of scientific and economic relevance, since potential findings may improve crop yield of this important plant.

Initially reading through the abstract, and throughout all the text, honestly, there was no cohesion in the text and it is difficult to comprehend what the authors meant; hence, English language correction is highly necessary in first place. Just the paper needs major English language polishing.

Abbreviations should be stated the first time they are encountered in the text, like in “According to positive MSE mode by”, nowhere in the text is what is MSE.

Line 53

          Sentence begins with “In our latest research …” and ends with no citation of the authors’ “latest research”.

Heading 2.2 “Acquisition of mass spectrum data”

The authors should state how QC samples were prepared, in more details.

Line 202

          of 3, minimum signal of 100000 to provide” very small/big values should be given in scientific format, i.e. 1e5, or 105.

Line 151

“In the study of Xie and others” should be “Xie et al”

Line 152

“acid, and lucidenic acid et al.,” maybe a typo

Round 2

Reviewer 2 Report

The text coherence is low. I suggest that the authors use professional English language editing. There are a lot of ambiguous sentences where is difficult to trace the meaning. 

Also, there should be references placed where appropriate to signify a claim. 
